# Nanomechanical Mapping of Hard Tissues by Atomic Force Microscopy: An Application to Cortical Bone

**DOI:** 10.3390/ma15217512

**Published:** 2022-10-26

**Authors:** Marco Bontempi, Francesca Salamanna, Rosario Capozza, Andrea Visani, Milena Fini, Alessandro Gambardella

**Affiliations:** 1Struttura Complessa Scienze e Tecnologie Chirurgiche, IRCCS Istituto Ortopedico Rizzoli, Via di Barbiano 1/10, 40136 Bologna, Italy; 2School of Engineering, Institute for Infrastructure and Environment, The University of Edinburgh, Thomas Bayes Road, Edinburgh EH9 3JL, UK

**Keywords:** cortical bone, nanomechanical mapping, atomic force microscopy, force mapping, tissue nanoindentation, nano biomechanics, hard tissues, elasticity, biomaterials

## Abstract

Force mapping of biological tissues via atomic force microscopy (AFM) probes the mechanical properties of samples within a given topography, revealing the interplay between tissue organization and nanometer-level composition. Despite considerable attention to soft biological samples, constructing elasticity maps on hard tissues is not routine for standard AFM equipment due to the difficulty of interpreting nanoindentation data in light of the available models of surface deformation. To tackle this issue, we proposed a protocol to construct elasticity maps of surfaces up to several GPa in moduli by AFM nanoindentation using standard experimental conditions (air operation, nanometrically sharp spherical tips, and cantilever stiffness below 30 N/m). We showed how to process both elastic and inelastic sample deformations simultaneously and independently and quantify the degree of elasticity of the sample to decide which regime is more suitable for moduli calculation. Afterwards, we used the frequency distributions of Young’s moduli to quantitatively assess differences between sample regions different for structure and composition, and to evaluate the presence of mechanical inhomogeneities. We tested our method on histological sections of sheep cortical bone, measuring the mechanical response of different osseous districts, and mapped the surface down to the single collagen fibril level.

## 1. Introduction

Force mapping via AFM is an elegant and comprehensive method to characterize quantitatively the spatial distribution of mechanical properties of biological tissues at micro/nano scale. Given a sample topography, typically elasticity maps (stiffness or Young’s modulus) can be constructed collecting the stiffness values obtained by AFM nanoindentation [1]. This technique is ideal to investigate biological samples from neurons [2], cells [3,4,5,6], microorganisms [7,8], and soft bone tissues down to single collagen fibrils [9,10,11,12], where its extreme force sensitivity enables the detection of moduli ranging from a few of Pa to several MPa. However, force mapping is not routine for hard tissues, where moduli up to tens of GPa can be measured, and AFM indentation is challenging due to the lack of both a reliable experimental and theoretical framework to validate, process, and interpret consistently the large amount of experimental data. Averaged moduli up to ~20 GPa obtained by AFM nanoindentation were used to compare different osseous districts on trabecular and/or cortical bone [13,14,15,16], or dentine [17]; however, no insights on the spatial distribution of mechanical heterogeneity were provided. To the best of our knowledge, only a pioneering study on bovine cortical bone performed nanomechanical mapping measuring tens of GPa in moduli [18]. Investigating the small-scale biomechanics of hard tissues is of primary importance to correlate structure and function when the development of regeneration/degeneration phenomena must be detailed in relation to healing, aging and, not least, the presence of biomaterials interfaces [19,20,21,22]. In these contexts, the richness of detail offered by the AFM topographic mode, when operated in tandem with force mapping, allows for, in principle, the quantitative comparison of local gradients of the mechanical response among different sample regions, as well as those within the same topographic area. 

After a brief review on the fundamentals of the technique, the most common issues encountered during force mapping of hard tissues will be pointed out, and possible solutions suggested. 

The AFM- nanoindentation procedure can be summarized as follows: the AFM tip is brought in contact with the surface by a piezo drive, causing the deflection of the cantilever (with spring constant *k*) and indentation of the surface; then, the tip is retracted from the surface. During this loading/unloading operation the deflection *d* of the cantilever and the force *F* experienced by the probe are measured, i.e., a Force-distance (*F-d*) curve is recorded (Figure 1a). *F*_ad_ represents the pull-off (or adhesion) force experienced during tip detachment; it is generally affected by the local roughness and considered unimportant for moduli calculation if much smaller than the maximum applied load *F_max_* [23]. *F-d* (loading and unloading) curves can be acquired point-by-point in form of a grid pattern within the topography of interest, and then converted into force curves containing the penetration depth *h* (*F-h* curves, Figure 1b). This operation is usually carried out via calibration on a reference sample much stiffer than the measured sample [23]. The Young’s modulus can be extracted by an appropriate fitting model from each *F-h* curve, so that a grid pattern of *F-h* curves acquired within a given sample topography will provide an elasticity map of the same region.

For an isotropic, homogeneous sample and infinitesimal indentations, i.e., *h* much smaller than the dimension of the indenter, elastic deformation of surface is often assumed, and the Young’s modulus is calculated using the contact Hertz model [23]. This approximation is common with soft biological matter, which is generally assumed to deform elastically when probed with forces smaller than ~10 nN; in addition, due to the aqueous environment surrounding biological matter, some samples exhibit velocity-dependent deformation (viscoelastic behavior). In this case, different loading rates should be probed [23,24]. The geometry of the tip-indenter (typically spherical, conical, cylindrical or pyramidal) also should be known, in that the Hertz model explicitly depends on a geometrical factor containing the effective contact area between the indenter and the surface [23,24,25]. 

On hard tissues, the large (0.1–10 µN) forces used to produce appreciable deformation of sample are expected to cause permanent (inelastic) deformation of samples [24], while viscoelastic effects are generally considered unimportant. Inelastic deformations can be described by the Oliver–Pharr contact model, which operates on the unloading *F-h* curve under the assumption of elastic-plastic regime [23,24,25,26]. Nevertheless, it is possible that the structural and compositional inhomogeneities typical of the hierarchical structure of hard tissues may cause different mechanical responses, depending on applied load and length scale investigated [18]. For example, when cortical bone was investigated, inelastic [12,13,14,18], or elastic [15,16,17], deformation of samples was assumed. 

It is worth underlining that the information on whether elastic or inelastic deformation is occurring at a given nanoindentation point is fully contained in the corresponding *F-h* curve, and the degree of elasticity (or plasticity) of a given surface sample can be assessed using the loading–unloading hysteresis of such curve [23]. In the following, the hysteresis of each single-point AFM indentation will be considered, and an elasticity index *η_el_* introduced and implemented into force mapping analysis. In this respect, for moduli calculation loading and unloading force curves will be considered separately, and Hertz and Oliver–Pharr models applied independently to provide corresponding maps of the same region. 

Furthermore, an important task regards the choice of the cantilever probe to be used for optimizing the signal-to-noise-ratio during the nanoindentation operation. For example, assuming infinitesimal elastic sample deformation, the formula *k* ~ 1.25 *ER*, valid for spherical indenter, provides the ideal value of *k* to measure the modulus *E* using an indenter of radius *R* [27]. Thus, with *E* = 10 GPa, if *R* = 100 nm one would have *k* = 1250 N/m -that is unrealistic- and *k* = 125 N/m with *R* = 10 nm. Cantilevers with stiffness much below 100 N/m would be desirable as they are cost-effective, while small-radius tips facilitate the attainment of nanometric resolution in topographic mode. However, despite the difficulty of indenting hard surfaces, when sharp tips are used, the condition of the small indentation may be difficult to fulfil. For example, on a ~ 30 GPa-stiff sample, *h_max_* ~ 40 nm was reached with *R* = 15 nm and *k* ~ 56 N/m [18]. In that case, based on finite-element analysis (FEA) simulations, inelastic deformation of sample was assumed and the Oliver–Pharr model used to fit the data. On the other hand, if elastic deformation is taken into account, the use of sharp tips may be problematic to extract quantitative data. For example, if *R* = 10 nm, the condition *h* << *R* would imply considering indentations smaller than 1 nm, which is unfeasible: the Hertz model cannot be used in such case. 

To overcome this problem, Kontomaris and co-authors have recently underlined the exquisitely geometrical nature of the problem, proposing an approximated Hertz formula, which differs from the original Hertz formula by a factor dependent on a wide range of *h/R* ratios [28], and appears suitable for cases where elastic deformation of samples takes place at indentation depths comparable with the dimension of the tip. 

In this work, a standard experimental AFM setup with spherical *R* = 10 nm tips and cantilever stiffness *k* < 30 N/m was used to characterize histological sections of cortical sheep bone, previously embedded in poly-methyl-methacrylate (PMMA). Topographic images and force curves were acquired, and spatial maps of *h_max_*, *η_el_* and *E* calculated according to both elastic and elastic-plastic regimes constructed. The maps are used to compare qualitatively and quantitatively the mechanical response of two distinct osseous districts, namely bone callus and native bone. Afterwards, force mapping at the level of single collagen fibrils is also illustrated. 

## 2. Materials and Methods

### 2.1. Samples Preparation

This study analyzed cortical bone samples from several studies on bone regeneration; all studies were approved by the local Ethical Committee of IRCCS Istituto Ortopedico Rizzoli (Protocol number: 29060, 14 November 2008) and then by the Italian Ministry of Health. Briefly, different bone scaffolds were implanted in crossbred (Bergamasca) female adult sheep (55 ± 10 kg b.w. and age 3.0 ± 0.5 years) metatarsi critical cortical defect and explanted at 6 months follow-up period [29]. Briefly, surgeries were performed in aseptic conditions. Sheep were placed on the left side and the right hind-limb was shaved and disinfected. The metatarsus shaft was exposed through a medial approach directly above the bone to reach the medial side. A 3.5 mm broad titanium dynamic compression plate with eight holes was contoured to the shaft and holes were drilled with a high-speed perforator: three distally and three proximally to the defect. A standardized 2 cm defect was created with an oscillating saw between the fourth and fifth screw hole, under constant irrigation, while preservation of soft tissues was obtained by using two retractors. The 2 cm segment was then removed, and, after implant insertion, the plate was fixed to the bone with 3.5 mm screws and the soft tissues were closed. At the explant, metatarsi were fixed in formaldehyde washed with distilled water and dehydrated with ethyl alcohol solutions at increasing concentrations (from ethyl alcohol 70% to ethyl alcohol 100%) at intervals of 24–48 h per solution. Samples were then infiltrated in methyl-methacrylate, according to a protocol described previously [20]. After reaching the full suspension of polymethylmethacrylate (PMMA), the samples were placed in the solution and oriented according to the requirements of cut. After polymerization, the blocks containing the samples were sectioned along a plane parallel to the long axis of the implant (EXAKT Cutting Systems, GmbH and Co., Norderstedt, Germany). From each sample, three 100–200 µm-thick sections were selected for AFM measurements. The selected section was attacked on a special microscope slide (Microscope Slides, 50 × 100 × 1.5 mm, EXAKT); the precise adherence of the section to the acrylic glass was performed with a cyanoacrylate glue which allows a complete adhesion in about 20–30 min. After 1 day, to have a homogeneous and uniform histological surface, the section was thinned with a grinding system (550, ATM GmbH, Mammelzen, Germany) using abrasive papers of different granulation (Struers), from 600 to 4000 grit, up to a thickness of (40 ± 10) µm. Finally, the section was treated with a polycrystalline diamond spray and polished automatically with a polishing system (Saphir 550 Grinding/Polishing System). 

### 2.2. AFM Force Spectroscopy and Mapping

AFM measurements were performed by a NT-MDT (Moscow, Russia) system equipped with an upright optical microscope. NSG30 tips (NT-MDT, Moscow, Russia) with resonant frequency around 250 kHz, operating in Contact mode were used. As declared by the manufacturer, the last 500 nm from the tip apex is cylindrical and the apex has curvature radius *R* = 10 nm [30]. Two cantilevers with stiffness *k* = 27.4 N/m and 22.6 N/m respectively were used for the measurements shown here. *k* was measured according to Sader [23], a script implemented into the acquisition software (NOVA, MT-MDT, Moscow, Russia). The cantilever’s deflection sensitivity was calibrated before each measurement from the hard-contact regime of *F-d* curves obtained from a clean and nanometrically flat silica slice (~80 GPa in stiffness). To check tip integrity, z-axis calibration before and after measurements was carried out on a TGS1 calibration grating (NT-MDT, Moscow, Russia; grid TGZ1 with height (21 ± 1) nm). Prepared bone samples were then mounted on the sample stage and characterized both topographically and by extraction of force curves. Two-dimensional arrays of *F-d* curves were acquired at randomly selected areas. Nanoindentation experiments were conducted in air and at room temperature, using the same tip as for AFM imaging. Displacement-controlled nanoindentation was carried out at a maximum load of 7.0 μN for the measurements of Section 3.1 and 4.0 μN for Section 3.2. The loading/unloading rate was varied between 0.15 and 0. 50 μms^−1^ causing no appreciable variations in calculated moduli, hence indicating that in this interval viscoelastic effects were minimized; all measurements were carried out at 0.25 μms^−1^ rate. All topographic images were taken at 256 × 256-pixel resolution; the maps of Section 3.1 collected 30 × 30 curves; the map of Section 3.2 collected 20 × 20 curves. 1000 points per curve were collected. Several survey curves were taken before starting maps acquisition to verify that *h_max_* fell in the range desired (see later). A possible contribution due to presence of residual PMMA (1–3 GPa in stiffness [31,32]) was also investigated. To this aim, the topographic phase signal, due to its sensitivity to the viscoelastic behavior of the polymer [33] was acquired on several non-overlapped regions, showing no significant phase contrast associated to moduli variations. Thus, at the spatial resolution investigated, namely 20.000 nm/30 ≈ 670 nm, the observed moduli variations can be considered essentially intrinsic to the investigated tissues. 

### 2.3. Elasticity Index and Young’s Modulus Calculation

Following Figure 1b, from each *F-h* curve an elasticity index *η_el_* = S_unload_/S_load_ can be extracted [23]. For a totally plastic sample *η_el_* = 0 while for a totally elastic sample *η_el_* = 1. 

For Young’s moduli calculation, loading and unloading curves can be processed separately. If fully elastic deformation occurs (*h_f_* = 0), no matter if loading or unloading curve is processed. In case of an isotropic, homogeneous sample and infinitesimal deformation (*h* << *R*), the Young’s modulus *E* can be calculated by the Hertz equation, which in the case of a spherical indenter is [23,24,25]: (1)$$F_{spherical}=\frac{4ER^{\frac{1}{2}}}{3\left(1-\upsilon^{2}\right)}h^{\frac{3}{2}}$$
where *ν* is the material Poisson ratio, that is 0.3 for bone [15,24]. The Equation (1) can be rewritten as [28]:(2)$${F^{\prime }}_{spherical}=\frac{4CER^{\frac{1}{2}}}{3\left(1-\upsilon^{2}\right)}h^{\frac{3}{2}}$$
where:(3)$$C=c_{1}+\overset{N}{\underset{M=2}{\sum }}\frac{3}{2M}c_{M}R^{\left(\frac{3}{2}-M\right)}h^{\left(M-\frac{3}{2}\;\right)}$$

The coefficients *c*_1_, …, *c_N_* depend on the *h/R* ratio and are provided for *h/R* ranging from about 0.04 to 5 [28]. It is demonstrated that the Equation (2) reduces to Equation (1) for *h << R*, whereas for *h >> R* it takes the linear form:(4)$$F_{cylinder}=\frac{2ER}{\left(1-\upsilon^{2}\right)}h.$$

The Equation (4) is the Hertz equation for indentation by a cylinder of radius *R*; the elastic compression using a rigid cylinder is indeed the approximate result of a deep (*h* >> *R*) indentation using a spherical indenter [28]. Note that, from the same authors, a more straightforward version of the Equation (2) has also been provided [25].

If *h_f_* > 0 the deformation is inelastic (Figure 1a,b); only the unloading curve is considered, and the Young’s modulus can be calculated by the Oliver–Pharr formula [26]:(5)$$E=\frac{\sqrt{\pi }}{2}\left(1-\nu^{2}\right)\frac{S}{\sqrt{A_{c}}}$$

*S* is the derivative at the inception (or inversion point) *h_max_*, and A_c_ is the projected area of the indenter at contact depth *h_c_* for a spherical indenter geometry (Figure 1c) [25,26], namely:A_c_ = π(2*Rh_c_* – *h_c_*^2^)(6)
where:(7)$$h_{c}=h_{max}-\varepsilon \frac{F_{max}}{S}$$

For a spherical indenter *ε* = 0.75 [26]. Note that, for large *h/R* ratios, one has *h_c_* = *h_max_*/2 [25].

### 2.4. Data Elaboration

#### 2.4.1. Processing and Analysis of F-d Curves

Indentation maps by AFM produce a large amount of data that must be processed in a reasonable time and accurately. To this end, different algorithms (implemented as a Python module [34], version 3.10, Van Rossum G., Drake F.L.; Create Space, Scotts Valley, CA, USA) can be assembled in different ways depending on the user’s computational needs (Figure 2).

The processing software consists of 5 subsystems. The first is responsible for reading the data, either as individual *F-d* curves, groups of curves, or maps. The second processes the *F-d* curves, aligning them, separating loading and unloading curves to generate the *F-h* curves. This subsystem also applies the Equation (2) on the loading curve and the Equation (5) on the unloading curve. The third is responsible for the management of maps. The fourth subsystem implements the models described in the previous section, performing uncertainty propagation evaluation to define the uncertainty associated with the result.

Finally, the fifth subsystem handles the saving of results to files and the creation of reports to be shown to the user.

From the computational point of view, the software implemented for map processing loads a file containing the accessory parameters (*R*, *ν*, *k*) necessary for the calculation of Young’s modulus and the file with the *F-d* curves including their coordinates on the analyzed surface. Once all the curves in the map are loaded, they are aligned so that the contact point is as close as possible to the origin of the *F-d* reference. At this point the cantilever load line is applied. The program calculates the horizontal distance between the *F-d* curve and the cantilever load line by determining the *h* value. Then, *F-h* curves are constructed. Afterwards, the software runs internal diagnostics to assess the correctness and consistency of the data. On each curve, the correlation coefficient of fitting, *r^2^*, is calculated together with the Δ*E/E* ratio obtained propagating the uncertainties on *R* and *ν*. The rejection criteria on *r^2^* and Δ*E/E* can be tailored according to the given experimental context; in the present study, for example, *r^2^* > 0.95 and Δ*E/E* < 30% were imposed. Indentations falling out of the interval 2 nm < *h_max_* < 40 nm were also rejected, as they are considered out of the range of validity of the models used. Indentations with *h_max_* ~ 3 nm are often associated to reverse position of the approach-retract parts of the F-d curve [35], causing typically large errors in the calculated moduli and frequent rejection. All the invalid data were marked as “nan” (not a number) and left blank in the corresponding map.

#### 2.4.2. Statistical Analysis

The Root-mean square roughness values reported were computed each on 10 non-overlapped regions. In order to test osseous districts distinct for structure and composition, an uncertainty of 10% on *ν* = 0.30 (commonly taken for bone) was propagated following indications from previous works [15,24]. An uncertainty of 10% on *R* = 10 nm was also assumed to account for possible discrepancy with manufacturer’s information. The distributions of *h_max_*, *η_el_* and Young’s moduli extracted from the corresponding maps were neither lognormal (*r^2^* < 0.8) nor normal, as confirmed by Shapiro–Wilk normality test at 0.05 level; this agrees with previous observations [10], operating by Origin Software was carried out on the distributions of *h_max_*, *ε_el_* and Young’s moduli. The Mann–Whitney test operated by Python software was used to compare the obtained distributions of moduli, hence regarded as non-parametric distributions of the moduli.

## 3. Results

### 3.1. Comparing Elasticity between Surface Micrographs 

Two regions of interest, namely native bone and bone callus were chosen on each histological section for assessing mechanical differences between osseous districts of different structure, composition and function. In Figure 3, representative 50 × 50 µm^2^ AFM images recorded on bone callus (upper raw) and native bone (lower raw) regions are shown along with the corresponding maps of *h_max_* and *η_el_* (left to right). In average, the root-mean-square roughness decreased from (153 ± 66) nm to (39 ± 23) nm from callus to native, while *F_ad_* was below 50 nN in both. Significantly, *h_max_* decreases from callus to native, as suggested by the prevalence of white or light grey in the corresponding color map, indicating reduced penetration and therefore higher stiffness on these zones. A limited correspondence between the position and orientation of certain topographic heights and the spatial distribution of features in the *h_max_* maps can be noted, as for example evidenced by the arrows in the Figure 3a,b.

The Young’s moduli maps obtained according to spherical-modified Hertz and Oliver–Pharr models are reported, together with the corresponding *η_el_* maps. (Figure 4). Qualitatively, an increase in moduli from callus to native is observed, as expected by the corresponding variation of hmax observed in the previous figure. Yet, from the *η_el_* maps one notes that both callus and native regions exhibit prevalently elastic character (prevalence of white in Figure 4c,f). To make these considerations quantitative, values corresponding to 25th, 50th and 75th percentiles of the non-parametric distributions of hmax, *η_el_* and Young’s moduli are reported (Table 1). 

In the example shown, many values exceeding 30 GPa were measured, associated to regions with *h_max_* near 2 nm (see Section 2.4.1). Although these values may still pass the quality check outlined before, their contribution to moduli distributions cannot be considered fully reliable and they have therefore been bolded in Table 1. Within this limitation, the observed increase in moduli from callus to native is statistically meaningful (*p* < 0.001); moreover, the evidence of a narrower distribution of values around the 50th percentile in native compared to callus in both the models is recognizable.

### 3.2. Force Mapping at Single Collagen Fibrils Level

The force mapping on a 2 × 2 µm2 large callus region is shown (Figure 5). The topography evidences fibrillar morphology covered with small particles—hydroxyapatite crystals-, as observed on cortical and bone surfaces investigated with AFM [24,36] (Figure 5a; average roughness 7.8 ± 3.2 nm, Fad = (45 ± 5) nN). The hydroxyapatite crystals seen measure 111 ± 21 nm, in agreement of particle sizes of previous findings [36,37,38]. The results of 20 × 20 grid patterns of indentations are also reported. Contrary to the maps of Figure 4, in this case the scale reduction reveals more clearly the presence of sub-regions with high ratio of rejected moduli. As usual, these values appear mostly associated to short indentation events, like in the upper part of Figure 5b, corresponding to a (prevalently elastic) sub-region with *η_el_* ≈ 1 (Figure 5c); nevertheless, some “deep” indentations also occurred. 

It is noticed that the correlation between the topographic features and the maps is here weak or absent, while the broad distributions of moduli in Figure 5f reflect the inhomogeneous character of the callus region, as highlighted before, even at this length scale. From Table 2, it is highlighted that this region exhibits prevalent inelastic character (median *η_el_* < 0.5) with markedly lower stiffness compared to the large-scale findings of the previous section. Note that here a lower indenting force was used (see Section 2.2 and Section 4.1).

## 4. Discussion

### 4.1. Generalities

This study aimed at developing and testing a method for multiscale characterization of hard tissues by means of force mapping based on AFM indentation, where the quantities relevant to the analysis (*h_max_*, *η_el_* and Young’s moduli in both elastic and elastic-plastic regimes) are obtained simultaneously from each *F-d* curve taken at the generic position on the surface and collected on a regular pattern to obtain a 2D map. Two important quantities should be set before starting the map acquisition, namely the maximum force load *F*_max_ and the spatial resolution of the map, or distance between the indents. If different regions of the sample with -in principle- different mechanical response and moduli are compared, as done in this work for large-scale regions of callus and native, indents at the same *F*_max_ should be operated for numerical comparisons. At the same time, *F*_max_ should determine that *h* falls in the range of validity of the model(s) used, that in this work is strictly between 2 and 40 nm to account for both the modified Hertz Equation (2) and Oliver–Pharr Equation (5). Note that this latter requirement may determine that the use of a unique *F*_max_ all over the surface gives inconsistent results, thus requiring to modulate its value according to local characteristics of the sample; of course, quantitative comparisons could be affected. 

It is underlined that the Equation (2) represents a helpful *escamotage* to fill the validity gap of the Hertz model in case of nanometrically sharp tips, for which indentation depths to be considered are comparable to or larger than the dimension of the tip. On the occasion, the Equation (4) (cylindrical shape) or other models, depending on the actual shape of the tip, can be used and elasticity maps constructed similarly when *h* >> *R*. Analogous considerations hold for the Equation (5). 

For what concerns the spatial resolution of the force mapping measurement, in case of inelastic deformation it is determined essentially by the sample deformation, originated by the overlapping of regions inelastically deformed or with residual stress, for which FEA simulations fixed the appropriate interindent spacing to ≈ 100 nm [18]. On the other hand, for purely elastic deformations the assumption of elastic half-space -at the basis of the Hertzian routes to extract the Young’s moduli; theoretical calculations suggested a minimum spacing of 5R [39]. It is clear that in the present context, where inelastic and elastic deformations are simultaneously considered, the “100 nm rule” was to be adopted during the acquisition of small-size maps like in Figure 5.

### 4.2. Elastic or Inelastic Sample Deformation?

The implementation of the elasticity index *η_el_* in force mapping adds additional information to the analysis in that it allows direct identification of regions characterized by permanent deformation (nonzero *h_f_* values). Basically, a look at its median value over the investigated region allows discriminates whether elastic or elastoplastic regime should be considered for further quantitative evaluations. In the study presented here, the slight tendency of callus to undergo permanent deformation compared to native is acceptable in light of its lower density and structural heterogeneity, typical of bone undergoing remodeling, while the higher degree of mineralization of the native led to smaller penetration depth, and hence increased stiffness, determining the observed different mechanical response of these two regions, where the modified Hertz rather than Oliver–Pharr model seems more suitable to fit the data as a consequence of having median *η_el_* close to unity. However, the observed heterogeneity implies that at small scales, local and distinct behaviors can be highlighted, such as the relevant plastic-like character of the smaller callus region in Figure 5. Moreover, reducing the space between indents exposes which points or regions are critical for nanoindentation, as shown by the white regions in the maps of Figure 5d,e. In this respect, the use of the more appropriate model to fit the data facilitates the analysis reducing the number of rejected values as can be easily observed by comparing the two maps. Clearly, the Oliver–Pharr, and not the Hertz model, may be considered to get reliable moduli estimation. However, it is underlined that direct comparison between values in Table 1 and Table 2 is not possible, mainly because they have been obtained at different values of *F_max_*. This is a possible reason for the sometimes large discrepancies between moduli obtained on similar samples, as evidenced particularly on bone [24]. Other reasons reflect the *relative* instead of *absolute* character of the mechanical stiffness obtained via AFM due to the basic assumptions of the used models [18].

Yet, it should be underlined that a deeper *a priori* characterization of the tip geometry also may contribute to improve the accuracy of the measurement. It is straightforward that such a characterization is beyond the scope of this paper, which did not focus on providing accurate numerical analyses of the sample investigated.

### 4.3. Heterogeneity of Samples and Other Applications

An important characteristic of force mapping is to provide a ready visualization of the possible interrelation between topographic image and the spatial distribution of the mechanical properties. In this respect, when dealing with the mechanical heterogeneity of cortical bone, a lack of correlation at the nanoscale has been highlighted [18]. In other words, the observed elasticity gradients are due to underlying local variations in structure and composition not resulting in a recognizable contrast at the topographic level. Other studies, although not corroborated by mapping, suggested that significant variations in mechanical properties could be attributed to variations in mineral content as well as to collagen fibril orientation and anisotropy [13,14,16,40]. In the cases shown here, despite the weak correlation observed at the single collagen fibril scale, the correspondence observed at the large scale between certain topographic features and certain maps features, as highlighted in Figure 3, may be originated by the collective view of a large number of weak small-scale correlations; such collective view may reveal an effective correlation originated by the presence of gradients of composition and structure that do have a topographic correspondence, but are visible only at large enough scales. In this sense, the force mapping operation can be regarded as a tool for revealing such gradients.

Although we focused on the measure of mechanical heterogeneity of bone as an ideal test bed, nothing prevents us from carrying out the considerations made here to other natural materials or biomaterials characterized by high stiffness. In this context, an important field of application is the presence of interfaces, that may be revealed by the corresponding near-surface (nano)mechanical variations of an extended number of parameters, namely *h_max_*_,_
*η_el_* and moduli. This approach, according to the guidelines depicted here, is advantageous when only a standard AFM setup is available, and when force mapping of samples cannot be performed by more complex modes of operation such as -for example, peak-force [41], or bimodal AFM [42].

As stressed here, once the geometry of the indenter is known with sufficient accuracy, *F_ad_* is negligible compared to *F_max_* and *F_max_* is chosen consistently, it is the shape of the *F-d* or *F-h* curve that determines the analysis.

## 5. Conclusions

This study established a comprehensive method for performing force mapping via AFM on hard (tens of GPa in stiffness) tissues using a standard experimental setup. We showed that the current limitations in the choice of the cantilever tip can be overcome adopting a model available in literature for describing the sample deformation comparable with the dimension of the indenter. Maps of maximum penetration depth, degree of elasticity of surface and Young’s moduli in both elastic and elastic-plastic regimes were calculated simultaneously on slices of cortical sheep bone within the 30 GPa range, allowing a quantitative comparison between the different spatial distribution of the mechanical properties between regions characterized by different structures and mineral content.

The method detailed here is potentially applicable to a broad class of hard materials and biomaterials.

## Figures and Tables

**Figure 1 materials-15-07512-f001:**
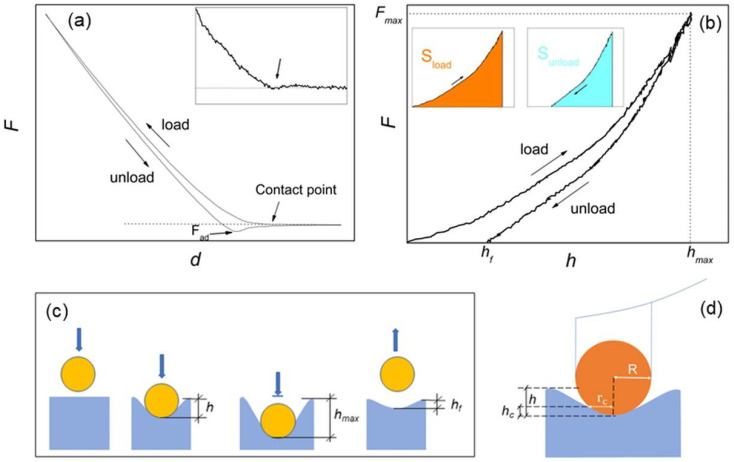
(**a**) force-displacement (*F-d*) curve with indication of the contact point (see also the inset) and adhesion force *F_ad_*; (**b**) force-penetration depth (*F-h*) curve equivalent to *F-d*, with indication of the effective (*h_f_*) and maximum (*h_max_*) penetration depths and the maximum loading force *F_max_*. In the inset, the colors evidence the areas under the load (S_load_) and unload (S_unload_) curves, respectively; (**c**) schematic of the spherical indentation, with graphical indication of *h*, *h_f_* and *h_max_*; and (**d**) compression by a rigid sphere of radius *R*, with indication of the contact radius *r_c_* and the contact depth *h_c_*.

**Figure 2 materials-15-07512-f002:**
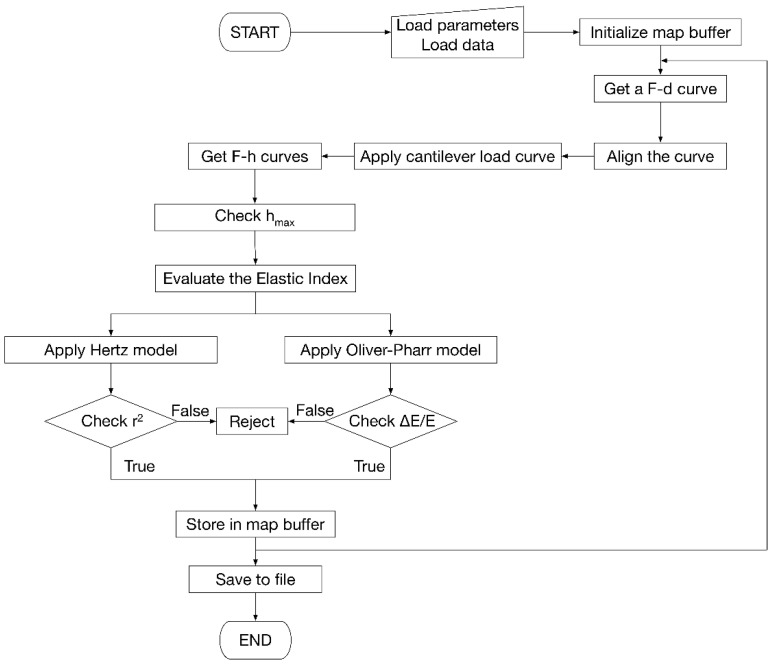
Flowchart of the algorithm developed for the evaluation of Young’s modules, including model selection criteria and map construction.

**Figure 3 materials-15-07512-f003:**
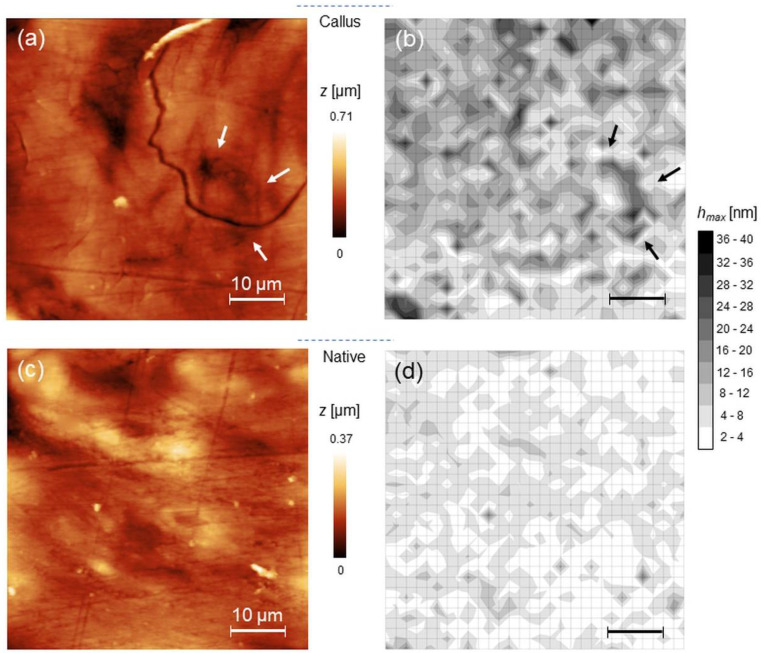
(**a**) representative 50 × 50 µm^2^ AFM topography of bone callus and the corresponding map of *h_max_* (**b**); (**c**,**d**) the same as (**a**,**b**) for native bone.

**Figure 4 materials-15-07512-f004:**
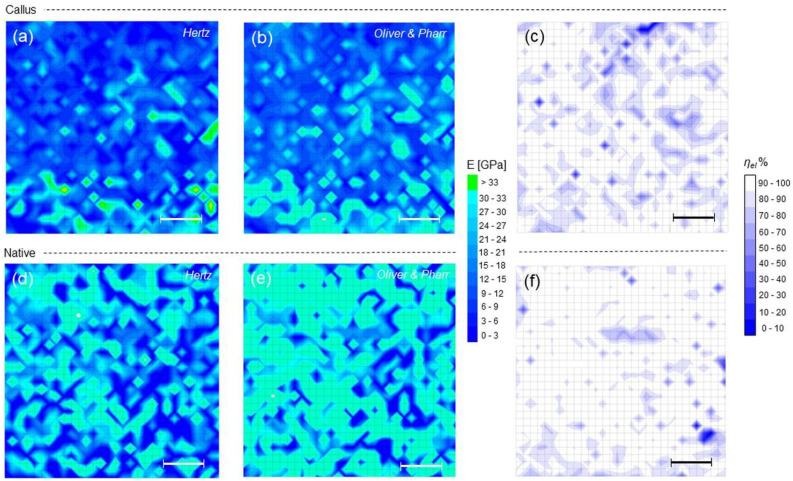
(**a**,**b**) Young’s modulus maps calculated according to Hertz and Oliver–Pharr models, and (**c**) *η_el_* map for callus; (**d**–**f**) the same as (**a**–**c**) for native bone.

**Figure 5 materials-15-07512-f005:**
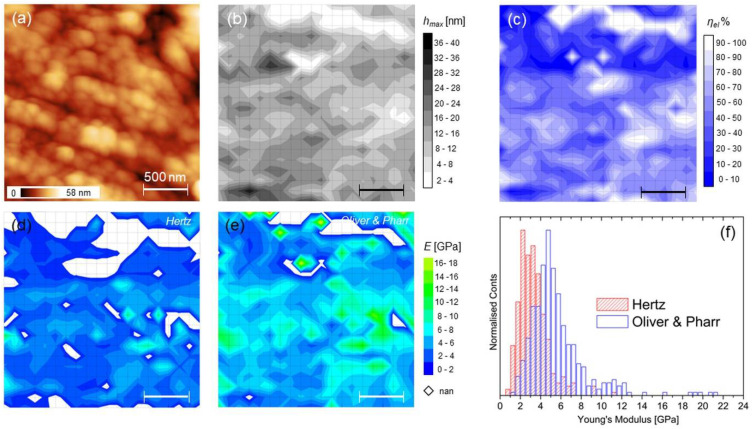
(**a**–**e**) The same as Figure 4 and Figure 5 for a 2 × 2 µm^2^ large area (callus region); “nan” regions in the moduli maps were left intentionally blank; and (**f**) Histograms reporting the moduli distributions according to Hertz (red bars) and Oliver–Pharr (blue bars) models.

**Table 1 materials-15-07512-t001:** *h_max_*, *η_el_* and Young’s moduli extracted from the maps of Figure 3 and Figure 4. The values span the 25th to the 75th percentile.

		Callus			Native	
	25th	50th	75th	25th	50th	75th
*h_max_* [nm]	8.90	12.8	17.2	4.58	6.01	7.70
0 < *η_el_* < 1	0.824	0.916	0.974	0.888	0.944	0.981
*E*, Hertz [GPa]	8.10	12.1	21.0	20.1	29.4	**40.0**
*E*, Oliver & Pharr [GPa]	9.60	14.7	24.8	26.3	**38.9**	**54.9**

**Table 2 materials-15-07512-t002:** The same as Table 1 for the 2 × 2 µm^2^ region of Figure 5.

	25th	50th	75th
*h_max_* [nm]	11.8	14.9	17.9
0 < *η_el_* < 1	0.331	0.488	0.663
*E*, Hertz [GPa]	2.36	3.04	3.88
*E*, Oliver–Pharr [GPa]	4.03	4.99	6.43

## Data Availability

Not applicable.

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
