# Peer review of "Nanomechanical Mapping of Hard Tissues by Atomic Force Microscopy: An Application to Cortical Bone"

_materials, 2022, doi:10.3390/ma15217512_

Round 1

Reviewer 1 Report

Sample preparation poses the most critical flaw of this research. Please check the reviewer report.

Author Response

Reviewer 1

This manuscript reports AFM mapping of elastic moduli on animal bone tissues using Hertz and OliverPharr model. Although this manuscript interesting integrates newly a developed formula from Kontomaris, the reviewer regrets to say this manuscript lacks in careful experimental design, which critically compromised the entire research. The reviewer feels obliged to recommend (for now) a rejection of this manuscript, and would like to point out the following concerns for future improvement.

We thank Reviewer 1 for Her/His valuable comments. In the following, a point-by-point reply to all the raised concerns is provided, hoping that this may help to reconsider his decision.

Major Concerns 1) Sample preparation: The process appears to follow the standard protocol of tissue paraffinization, where a biological tissue is covalently crosslinked with PFA, dehydrated with EtOH gradient, and embedded in paraffin (in which the authors only described as “polymerization”). If paraffinization was indeed done, the measured maps of elastic moduli would reflect paraffin rather than the tissue itself. Poisson ratio of 0.3 would be invalid and thus require the reports of indentation moduli (which was absent in this manuscript). In contrast, ref.18 (which the authors regarded as the only study that “…constructed elasticity maps on bovine cortical bone”) clearly stated that “sample preparation did not involve chemical treatment, ethanol dehydration or embedding.” This concern is the most critical part to the manuscript, which the authors must either justify with reasonable explanation or else with new data collection entirely.

Actually, we did not mention parafinization, as Polymethylmetacrylate (PMMA) was used for polymerization, as specified in the Introduction (Former line 127, now line138) and Materials and Methods (Former lines 177-179, now lines 168-171). To improve the clarity on this point, we have now added some experimental details and Ref. 20, where the embedding procedure is extensively described, is now better placed in the M & M section. Other details have now been added as information on animals (Lines153-157). For Ref. 18, the very high purity of the bone surfaces was necessary requirement to highlight the bone’s intrinsic heterogeneity, which is not the main aim of our work (see later); anyway, hard-resin embedding followed by cutting in slices and careful grinding/polishing operation provides a clean surface and relatively flat surface, providing a useful bed test for AFM analyses.

Actually, we regard to Ref. 18 as the work that provided maps of bone with up to 30GPa in moduli. We understand that this sentence could be misleading, so we have slightly modified it (lines 46-48).

2) Mechanical Model: This manuscript states in Line 160-162 that “the last 500 nm from the tip apex is cylindrical”. However, ref.29 which the authors specifically cited states that “the last 500nm from tip apex is conical” (which is quite obvious from the source’s illustration). This makes the approximation with Equation (4) invalid, particular for those data points on bone callus with hmax > 10 nm tip radius.

I regret to verify that the reported sentence from the manufacturer is just “the last 500 nm from the tip apex is cylindrical”, as stated in the manuscript (please see screenshot attached). Please note that 500 nm is about 3% of the whole height of the tip (15 um, manufacturer), thus in the image the indicated 500 nm are referred to the very last part of the tip, not visible in the zoomed-in image.

3) Cantilever choice: NSG30 cantilever was used to for the nanoindentation experiment. However, according the manufacturer, the cantilever was designed for tapping mode (intermittent and noncontact) - https://ntmdt.nl/product/nsg30/ and https://www.ntmdt-tips.com/products/view/nsg30 . Moreover, the spring constant and resonant frequency of the tips used in the experiment were within the lower range of the typical values reported by NT-MDT. Although the authors justified their choice of soft cantilever on Line 108-110, the reviewer would like to raise the possibility that using soft tip on hard samples tends to permanently destroy the tip surface or permanently bend the cantilever as well. From the reviewer’s experience, cantilever bending is the most problematic in 256 x 256 mapping.

NSG30 can be used in tapping mode for imaging on “hard” samples due to its relatively high k compared to, for example, NSG10 type, however, nothing prevents from using NSG30 for contact and nanoindentation measurements as we did, as this improved greatly the image quality compared to tapping with the same tips. 20-30 N/m-stiff cantilevers are not properly soft; unfortunately, the actual k and resonant frequency values differ a lot from what specified by the manufacturer, so using the lower range respect to the typical values is not a choice (on the box, the range 22-100 N/m is indicated, but in our experience none of the cantilevers exceed 35 N/m). As concerns permanent bending, some precautions can be taken to avoid it eventually during calibration (see later). If the Reviewer means 256 x 256 as mapping resolution, we underline that it would be critical because of the huge time required for acquisition and too small space between indents. That is because we used 30 x 30 or 20 x 20 grid patterns for affordable measurements.

4) Calibration: The author’s choice to do Sader calibration is good to preserve the tip integrity. However, the reviewer does not understand why the authors still need to do contact calibration of deflection sensitivity with 80GPa silica, which poses a high risk of permanent bending to the cantilever. Sader’s method should already provide the amplitude inverse optical lever sensitivity (Amp InvOLS) which is equal to Defl InvOLS multiplied with ~1.09. see https://aip.scitation.org/doi/10.1063/1.2162455 5)

The test implemented into NOVA program calculates the cantilever’s k given its dimensions, the measured resonant frequency and Q factor, as application of the Sader’s formula. The Higgins method that the Reviewer signals deserves to be explored in our case, and we will implement it surely, however, this method would require careful validation in our own experimental and, as consequence, additional and dedicated work compared to the well consolidated and widely used contact calibration. On the other hand, the latter can be carried out avoiding permanent damages to tip/cantilever if parameters (Lift/Land distances and speed) are selected carefully. The tip integrity was checked before and after measurement on TG-1 calibration grating and k measured very often during operation, as now highlighted (lines 195-197).

Objective: The authors state multiple objectives: line 19-20 said “we propose a protocol to construct elasticity maps ….”; line 360-361 said “this study aimed at developing and testing a method for multiscale characterization of hard tissues…”; line 366-367 said “As the objective of this work is to compare two regions (callus and native) …” However, the results only report the mechanical measurement of bone callus and native bone, without any protocol/method verification or description on analysis development. Since many researchers already perform force mapping on bone, the authors showed be clearer on why this result is significant or important or what scientific questions they answer.

We thank the Reviewer for raising this issue; we have now modified the sentences indicated to better clarify the objectives of our work (what exposed at lines 428 and following repeats develops what anticipated in the abstract, while (see lines from 434 on) the remaining has been rephrased.  

Minor Concerns 1) At least two force-distance curves (corresponding to Hertz and Oliver-Pharr model each) should be given with clear indication of the zero-contact point.

An inset in Fig. 1a has now been added, providing the required zoomed-in curve. Actually, at each nanoindentation point there is just one curve (F-d, transformed into F-h, then load and unload curves processed separately to obtain H and OP moduli).

2) Percentile representation of data is quite inappropriate since the data obviously seems to follow lognormal distribution. If the authors want to show data heterogeneity, the reviewer recommends showing histogram distribution and quantified with mode or distribution fitting values.

Log-normal fittings on our data provided low (<0.8) r2; this circumstance (neither gaussian nor lognormal distributions of moduli) was already observed on cortical bone (Ref. 10) and is not surprising due to the heterogeneity of such samples. Thus, following the literature we reported percentile representation as more appropriate in this context. However, we remark that our purpose was not to provide exact estimations of the bone stiffness, but to develop a method to extract nanomechanical mapping from these surfaces.  

Reviewer 2 Report

In the manuscript entitled “Nanomechanical Mapping of Hard Tissues by Atomic Force Microscopy: An Application to Cortical Bone”, by Marco Bontempi et al., the Authors presented the mechanical comparison between native bone tissue and bone callus. The authors presented an original approach that derivate from previous publications. It allows for simultaneously calculating Young’s modulus of investigated samples and the determination of the plastic-elastic regime. The method allows not only to show a (mean) value but provides insight into the mechanical heterogeneity based on elasticity maps. The advantage of the model is that it can be implemented in any AFM and does not require novel modes such as QI, or PeakForce etc.

The article is presented in a well-organized way, supported with good-quality graphics. The introduction is well written guiding the reader to the topic of the research on the elasticity of hard samples. (1) The only missing part is the paragraph describing the biology of bone callus highlighting the importance of the presented research.

The methodology is extensively presented, which is an advantage. (2) Again, the missing part of the biological origin of samples is a drawback. The bone callus varies in stiffness with time from its formation. There is a transition from soft callus to stiff callus within weeks of its formation. The precise data on the time frame from its formation should be presented. There is a statement of Ethics, but the information about animals (sex, age, number of animals used, number of samples tested), and biological repetitions are used to produce the data. At least the Authors should indicate an article where the reader can find additional biological data.

(3) My next concern is the grinding and polishing. In Fig.4 one can easily notice the lines originating from grinding. In some articles, only fractured samples were investigated (Ref35). I noticed that this procedure is successfully used in the literature, but I would like to ask the Authors to comment on that why such an approach was selected. Maybe the use of the reference that the sample topography is not influencing the nanomechanical properties of the stiff samples (ref 18).

The Authors show that the advantage of their method is imaging and nanomechanical measurements in the same area. Did the Authors try to measure the same area in contact mode after indentation? I wonder if using a sharp tip is possible to visualize the plastic deformation after scanning. It is just a comment, but in my opinion, it would create direct proof of plastic deformation.

(4) The discussion is a part that in my opinion the weakest part of the manuscript. After finalizing the reading I was not sure which method the Authors consider being better: Hertz or Oliver & Pharr? (5). It should be natural to select the second, but the data in Table 1 suggest lower applicability of it (bold) . A paragraph describing the relevance of this model to other stiff materials would be valuable; or an addition of future perspective; biological relevance.

(5) The discussion of the results in my opinion lack the identification of specific areas of a sample that are biologically relevant. The distribution of Young's modulus is of course clear from the images – the mechanical heterogeneity is presented, but there is no (try of) explanation why the Authors observed such differences. I mean what is the point of showing a map of callus rather than presenting a mean Young’s modulus extracted from the map? If the Authors could present literature or different methods showing that heterogeneity could be an issue (biologically) it would significantly increase the output.

Minor/editorial

I found in line 160: “resonant frequency 256 kHz” while on the manufacturers website there is 320 kHz.

How long one cantilever was used? Did the Authors determine its shape after the measurement? Using such a large loading force can blunt the tip quickly. I would recommend using TGT-1 grating from NT-MDT before and after the measurement.

I could not find an explanation what is the difference between hc and hf in the equations and Fig1?

Line 271: "the rejection criteria r2 > 0.95 and ΔE/E < 30% were used". I would put these criteria in Fig2 for clarity.

Author Response

Reviewer 2

In the manuscript entitled “Nanomechanical Mapping of Hard Tissues by Atomic Force Microscopy: An Application to Cortical Bone”, by Marco Bontempi et al., the Authors presented the mechanical comparison between native bone tissue and bone callus. The authors presented an original approach that derivate from previous publications. It allows for simultaneously calculating Young’s modulus of investigated samples and the determination of the plastic-elastic regime. The method allows not only to show a (mean) value but provides insight into the mechanical heterogeneity based on elasticity maps. The advantage of the model is that it can be implemented in any AFM and does not require novel modes such as QI, or PeakForce etc. The article is presented in a well-organized way, supported with good-quality graphics. The introduction is well written guiding the reader to the topic of the research on the elasticity of hard samples.

We thank Reviewer 2 for carefully reading our manuscript and raising very interesting concerns, for which we have attempted to reply at our best, also improving some parts of the main text.

(1) The only missing part is the paragraph describing the biology of bone callus highlighting the importance of the presented research. The methodology is extensively presented, which is an advantage. (2) Again, the missing part of the biological origin of samples is a drawback. The bone callus varies in stiffness with time from its formation. There is a transition from soft callus to stiff callus within weeks of its formation. The precise data on the time frame from its formation should be presented. There is a statement of Ethics, but the information about animals (sex, age, number of animals used, number of samples tested), and biological repetitions are used to produce the data. At least the Authors should indicate an article where the reader can find additional biological data.

We thank the Reviewer for raising this issue. As underlined, the bone callus varies in stiffness with time from its formation and there is a transition from soft callus to stiff callus within weeks of its formation. Literature data showed that animals such as sheep have bone structures and remodelling processes comparable to those of humans and are thus recommended for studies of clinical fixation strategies and mechanical regulation of repair processes. We have now added further statements on the used samples and on the hard resin embedding procedure throughout section 2.1.

(3) My next concern is the grinding and polishing. In Fig.4 one can easily notice the lines originating from grinding. In some articles, only fractured samples were investigated (Ref35). I noticed that this procedure is successfully used in the literature, but I would like to ask the Authors to comment on that why such an approach was selected. Maybe the use of the reference that the sample topography is not influencing the nanomechanical properties of the stiff samples (ref 18).

Grinding/polishing are carried out after cutting to provide an as possible as flat surface for histological observations to the optical microscope, also removing residual PMMA content as much as possible, exposing the bare bone surface (Ref. 20). We suggest that topography may affect the mechanical properties at large scales, while at small scales none or weak relation between topography and stiffness was found; in this respect, this agrees with Ref. 18 that at small (2 µm) scales, where contrasts in the nanomechanical maps of stiffness were attributed to “…underlying gradients in structure and composition…”, rather than to topographic features.

The Authors show that the advantage of their method is imaging and nanomechanical measurements in the same area. Did the Authors try to measure the same area in contact mode after indentation? I wonder if using a sharp tip is possible to visualize the plastic deformation after scanning. It is just a comment, but in my opinion, it would create direct proof of plastic deformation.

In principle the answer is “Yes”, an additional topography could test whether a permanent deformation was actually produced; however, small scales would be interesting prevalently, because the mark produced by the  tip during indentation is very small (indentations are in average only a few nm in depth, and the contact area with sharp tips is of the order or the 2πR2, hence about 600 nm2 in our case. This is 1/10000 with a 50 µm-large topography and 1/10 with a 2-µm large one). However, we believe that the elasticity index (now rewritten as ηel) provides a good indication on whether inelastic deformation occurred, facilitating the analysis.

(4) The discussion is a part that in my opinion the weakest part of the manuscript. After finalizing the reading I was not sure which method the Authors consider being better: Hertz or Oliver & Pharr? (5). It should be natural to select the second, but the data in Table 1 suggest lower applicability of it (bold) . A paragraph describing the relevance of this model to other stiff materials would be valuable; or an addition of future perspective; biological relevance.

We thank the Reviewer for helping us to improve this part of the manuscript. Now the Discussion has been split into parts, which should improve the clarity and readability of the manuscript and better clarify the purpose of the study.

As mentioned above, we introduced ηel to establish whether the surface is better described by an elastic or elastic-plastic model. In fact, the lower applicability of OP in native compared to H is likely due to the prevalent elastic character of the surface, which lead to a misuse of OP in this case (too high values). However, the situation is complicated by the expected dependence on the applied load (Fmax), in that the higher the force, the higher the possibility of causing permanent deformation, so it is very difficult to drag general conclusions from these measurements. Alle these considerations have now better highlighted (Please see the new Discussion section).

(5) The discussion of the results in my opinion lack the identification of specific areas of a sample that are biologically relevant. The distribution of Young's modulus is of course clear from the images – the mechanical heterogeneity is presented, but there is no (try of) explanation why the Authors observed such differences. I mean what is the point of showing a map of callus rather than presenting a mean Young’s modulus extracted from the map? If the Authors could present literature or different methods showing that heterogeneity could be an issue (biologically) it would significantly increase the output.

We thank the reviewer for raising this issue. First of all, it must be emphasised that we do not intend to provide a tool for the detailed characterisation of bone with its heterogeneity, and the explanation of why certain details were observed and/or are biologically relevant would require efforts beyond the purposes of our work. In this sense the heterogeneity of bone is, in our case, just a good test bed for the technique, for which also extracting distributions of values from the same maps is an added value of the analyses. Nevertheless, to answer the Reviewer's question, it must be kept in mind that a variety of situations may arise in which the ability to spatially distinguish zones of different stiffness (or different elasticity index, or other related property) is of crucial importance; this applies, typically, with material-bone interfaces. An example is the presence of a biomaterial (a biocompatible ceramic, for example) mixed with the bone matrix for purposes of regeneration, where force mapping may distinguish regions that exhibit different mechanical behaviours as a consequence of gradients of composition and/or structure, or also as consequence of the development of regeneration processes. In these contexts, mapping adds information compared to providing mean Young’s moduli. Actually, this is our future application of the method introduced in the present manuscript. Changes have now been made in the manuscript to account for the considerations above.

Minor/editorial

I found in line 160: “resonant frequency 256 kHz” while on the manufacturers website there is 320 kHz.

Actually, two cantilevers were used with frequencies around 260 kHz, within the range indicated on the NSG30 box (240 – 440 kHz). The 320 kHz of the website is an indicative average value.

How long one cantilever was used? Did the Authors determine its shape after the measurement? Using such a large loading force can blunt the tip quickly. I would recommend using TGT-1 grating from NT-MDT before and after the measurement.

Tips/cantilevers integrity was checked using NT-MDT - TGS-1 grating (on the 21 +/- 1 nm- pattern) before and after the measurement; blunt tips, actually seldom found, were promptly replaced (please see Section 2.2).

I could not find an explanation what is the difference between hc and hf in the equations and Fig1?

Fig 1 and the theoretical part have been improved and the meaning of hf, hmax and the other relevant quantities underlined more clearly. Please check Fig. 1 caption and the partially rewritten Section 2.3.

Line 271: "the rejection criteria r2 > 0.95 and ΔE/E < 30% were used". I would put these criteria in Fig2 for clarity.

Fig. 2 is referred to a general case, so we would prefer to keep the indication of having used the “r2 > 0.95 and ΔE/E < 30%” criteria specifically for the measurements shown here in the text; however, we have now slightly modified those sentences to improve the clarity (lines 310-320).

Reviewer 3 Report

The authors report on new methods to be used when indenting hard biological samples using classical AFM indenters. They review different methods and conclude that elastic and inelastic deformations are better represented by spherical modified Hertz and Oliver-Pharr's models. Interesting results on sheep cortical bone are shown and elastic maps are presented, using callus and native bones. Heterogeneous structures, due to the compositions in hydroxyapatite and collagen concentrations are featured.
This is a nice study depicting how to solve these difficulties when hard materials are involved, and the paper reads very well. The methodology and results are convincing. I think that the paper is worth to be published because it presents a new way to determine hard materials behavior using AFM, at nanometer resolution, relevant to materials subtle changes .

A small question raises when reading the part on "force mapping at collagen fibrils level". It is true that, at the scale used 2µm x 2µm, one can map the structure and find differences betwwen hydroxyapatite and collagen parts. A better discussion could rely on the values found and how relevant they are, with respect to these two components. What does the litterature data tell us ? This could be a point for the discussion, since here only focus is made on the measurement scale, but no numbers are provided. For example, is 2 GPa a correct value for collagen ?

MINOR DETAILS
- page 4: h/R ratios, the ... (1) can be rewritten => missing "equation" ??
- page 5: Equation (4) is Hertz formula

Author Response

Reviewer 3

The authors report on new methods to be used when indenting hard biological samples using classical AFM indenters. They review different methods and conclude that elastic and inelastic deformations are better represented by spherical modified Hertz and Oliver-Pharr's models. Interesting results on sheep cortical bone are shown and elastic maps are presented, using callus and native bones. Heterogeneous structures, due to the compositions in hydroxyapatite and collagen concentrations are featured.
This is a nice study depicting how to solve these difficulties when hard materials are involved, and the paper reads very well. The methodology and results are convincing. I think that the paper is worth to be published because it presents a new way to determine hard materials behavior using AFM, at nanometer resolution, relevant to materials subtle changes.

We thank Reviewer 3 for appreciating our work. In the following, we reply to the raised concerns at our best.

A small question raises when reading the part on "force mapping at collagen fibrils level". It is true that, at the scale used 2µm x 2µm, one can map the structure and find differences between hydroxyapatite and collagen parts. A better discussion could rely on the values found and how relevant they are, with respect to these two components. What does the litterature data tell us? This could be a point for the discussion, since here only focus is made on the measurement scale, but no numbers are provided. For example, is 2 GPa a correct value for collagen?

Actually we provided the values extracted from the maps of the 2 x 2 region in Table 2. For collagen with uniform coating of hydroxyapatite platelets, stiffness values are in the range of GPa [Ref. 24]. However, besides a lack of studies concerning the mechanical of cortical sheep bone at the length scale of our study, a range of variability of stiffness is currently found even on similar samples, in that every measure is influenced mainly by the load applied and, secondarily, by other factors like the geometry of the tip-indenter, the used model, etc (this fact has been pointed out at lines 480-487).

MINOR DETAILS
- page 4: h/R ratios, the ... (1) can be rewritten => missing "equation" ??
- page 5: Equation (4) is Hertz formula

These items have been corrected. All the theoretical part has now been improved (please see Section 2.3).  

Reviewer 4 Report

This manuscript deals with a protocol to construct elasticity maps of surfaces up to several GPa in moduli by AFM nanoindentation using standard experiment conditions. There are no particular problems with the content of this manuscript, but there are some minor parts that need confirmation or revision before publication of the paper.

1. Figure 3 

I don't understand why Fig. 3 is essential to explain the Python module in the experiment. Since there is not much information in Fig. 3 and the manuscript becomes redundant, I think that it's better to explain concisely in the main text without Fig. 3.

2. Figure 6a

The validity of Fig.6 a is claimed by citing the papers [24, 34], but it looks like an AFM image measured with a relatively blunt tip. In order to prove the author's claim, is it possible to show an AFM image of the same surface measured with another sharp tip?

3. Figure 6c-e

The author explained the reason why the "nan" regions in Fig. 6c-e are generated as "Contrary to .... calculated Young's modulus." in the main text. Could you please explain why the calculation at these regions are not possible? In particular, I would like to know a clear reason why the author cannot calculate the elasticity even by using Hertz model?

Author Response

Reviewer 4

This manuscript deals with a protocol to construct elasticity maps of surfaces up to several GPa in moduli by AFM nanoindentation using standard experiment conditions. There are no particular problems with the content of this manuscript, but there are some minor parts that need confirmation or revision before publication of the paper.

We thank Reviewer 4 for appreciating our work. In the following, we reply to the raised concerns at our best:

  1. Figure 3 

I don't understand why Fig. 3 is essential to explain the Python module in the experiment. Since there is not much information in Fig. 3 and the manuscript becomes redundant, I think that it's better to explain concisely in the main text without Fig. 3.

We agree with the Reviewer that Fig. 3 is somewhat redundant, so we have removed it and modified the relative section 3.4 accordingly.

  1. Figure 6a

The validity of Fig.6 a is claimed by citing the papers [24, 34], but it looks like an AFM image measured with a relatively blunt tip. In order to prove the author's claim, is it possible to show an AFM image of the same surface measured with another sharp tip?

We thank Reviewer 3 for this observation. Measuring collagen clearly is quite challenging in air conditions, so we have other images but due to the presence of hydroxyapatite platelets which uniformly coat the fibril, the latter cannot be distinguished so clearly as like in the image provided. Despite of dragging conclusions of the heterogeneity of the nano mechanics of cortical bone, in Ref. 18, to which we mostly refer for nanomechanical mapping, single collagen fibrils are barely visible. Nevertheless, we list some reasons to consider Fig. 6a as genuine, hoping this may lead the Reviewer to reconsider Her/His observation.

  • We referred in particular to Ref. 36 (former Ref. 34) as it reports an AFM measurement very similar to ours in a very similar case; Collagen fibrils coated with hydroxyapatite crystals have been observed in a magnitude of studies utilizing AFM, among other techniques, with very similar results.
  • Prior to measurement, the tip sharpness was checked on a calibration grating (see M & M section).
  • There is reproducibility in the average size of the hydroxyapatite platelets coating the fibril respect to literature (around 110 nm).

  1. Figure 6c-e

The author explained the reason why the "nan" regions in Fig. 6c-e are generated as "Contrary to .... calculated Young's modulus." in the main text. Could you please explain why the calculation at these regions are not possible? In particular, I would like to know a clear reason why the author cannot calculate the elasticity even by using Hertz model?

We thank the Reviewer for raising this issue, which gives us the opportunity to better clarify this aspect of our work. H or OP model are both implemented in this work; however, depending on the prevalence of elastic or inelastic behaviour (the measure of which is provided by the elasticity index, now named ηel), the one or the other model may be more appropriate in describing the mechanics of the surface. For example, with H and OP maps of former Fig. 6 (now Fig. 5), the prevalent inelastic character of the surface suggests that OP model is more suitable than H to fit the data. This is -in turn- evidenced by the higher occurrence of “nan” in the H map compared to OP map. In this new version of the manuscript, we have better clarified the concept of “nan” and why calculation is not possible in some regions. “nan” means essentially that the value was rejected by the quality check of the Python routine, which happens for the following reasons:

  • hmax > 40 nm (surface “too soft” for the used Fmax, needs to implement models (both H and OP) for h >> R)
  • hmax < 2 nm (surface “too hard” for the used Fmax, no value provided at all (Ref. 25)). If hmax ~ 2-3 nm, this is often associated to the “curve reverse” phenomenon described in Ref. 35; if reverse is checked, the algorithm goes further on to check whether r2 of the Hertz model and ΔE/E of the OP model are ok; if Yes, an output value is provided as well.
  • “bad” r2 of the Hertz model (only OP model in output)
  • “bad” ΔE/E of the OP model (only H model in output)
  • reasons 3 + 4 (no value provided at all).

The occurrence of conditions 3), 4) or 3 + 4) is typically associated to noisy curve, which do not pass the tests. Surely, there can be points of the surface where the used models are not valid; in this case, yet the conditions (3) and/or (4) lead to reject the datum. All these considerations have now been summarized better in Section 2.4.

Round 2

Reviewer 1 Report

Thank you for the revision. Please see the attached reviewer report in details.

Author Response

Thank you for the revision. Now the reviewer understands this study purpose, which is “to provide a method to extract nanomechanical mapping from these surfaces”, not “to provide exact estimations of the bone stiffness”.

We thank the Reviewer for helping us to improve the clarity of our manuscript as concerns the purposes of our study. However, I underline that the sentence “to provide exact estimations of the bone stiffness” was never written in our manuscript. To cite this Reviewer, we actually wrote (in the previous version):

line 19-20 said “we propose a protocol to construct elasticity maps ….”; line 360-361 said “this study aimed at developing and testing a method for multiscale characterization of hard tissues…”; line 366-367 said “As the objective of this work is to compare two regions (callus and native) …

So, there is no mention of providing “exact” values. Instead, in the previous version we also wrote that our work:

“[….]  focuses on the development of a method for a comprehensive characterization of hard tissues rather  than providing accurate numerical analyses of the sample investigated”.

 In this case, the reviewer would like to share the following perspective. • The method to exact nanomechanical mapping already existed in Asylum Research’s AFM module (where users can apply Hertz/Sneddon, Oliver-Phar, JKR, DMT). Please see https://afm.oxinst.com/assets/uploads/products/asylum/documents/Nanomech-Pro-NanomechanicalAFM-Techniques-for-Diverse-Materials.pdf. That means, the novelty of “providing a method to extract nanomechanical mapping” is low.

We provide a method, as equations are taken from new and recent (Kontomaris’) works. The Asylum’s algorithm implements “classical” models which could not be suitable for mapping reliably hard tissues, because of the reasons discussed extensively in the Introduction and M & M of our work. So, there is novelty, unless the Asylum’s algorithm also calculates maximum penetration depth and elasticity index and implements Young’s moduli calculation when the penetration depth is comparable with the dimension of the indenter.  Instead, I read from the website that actually Contact Resonance Viscoelastic Mapping (therefore, a different technique) is used for providing “quantitative modulus of materials such as wood and bones”, as well as “Local mechanical characterization of high stiffness materials” (https://afm.oxinst.com/application-detail/afm-for-nanomechanical-measurements). Thus, the formulae implemented in simple Force-mapping are for soft samples characterization (biological tissues, polymers, hydrogels, as specified at the same webpage).  

  • However, the application of Kontomaris method in nanomechanical mapping is new, and should be emphasized or explained more clearly. In fact, the reviewer cannot access Ref.28 and have to rely on other papers by Kontamaris to understand the correction factor C and the Cm table. The reviewer thinks the authors could emphasize more on this point.

We have provided numerical details in the Section 2.3, however we did not developed this theory on our own, so instead of repeating Kontomaris’ calculations, or reporting the entire Table of coefficients of Ref. 28, we explained the generalities of the theory and mention explicitly these recently published works; we deem this is the correct approach to follow especially with a recent research to which one directly refers. Nevertheless, a more straightforward formula is provided in Ref. 25; a specific indication for this is now provided (line 244).

  • The author might need to reconsider the title since this work is not about mapping bone stiffness by AFM, but rather about method development for nanomechanical mapping. The current title is misleading.

We thank for this suggestion, but we have a different opinion as concerns the current title. It is divided into two parts:

 “Nanomechanical mapping of hard tissues via AFM”. The nanomechanical mapping focuses on hard tissues, because this is the most relevant application of the method developed, although hard polymers, or other biomaterials up to a few tens of GPa could be exploited, as well specified in the Discussion section.  

“An application to cortical bone”. Which, in our opinion, explains well what we actually did in our work. Note that It is “An application to”, rather than “An exact measure of”.

Thank you for clarifying the process and sorry for the misunderstanding. The reviewer understands that the main aim of this work is not to estimate the “exact value” of bone stiffness. But the reviewer would like the authors to improve the justification of 0.3 Poisson ratio (defined as side-ward expansion divided by axial compression) because PMMA might penetrate through the tissue fiber and distort the sample’s mechanical property. Theoretically, PMMA embedding would limit the side-ward expansion of bone tissue during compression or tip penetration, thus altering the Poisson ratio.

Poisson ratio = 0.3 is commonly taken for bone, however we have propagated a 10% uncertainty on this value to account reliably for local fluctuations due to heterogeneity, as indicated in the M & M section. The presence of PMMA was checked by phase contrast measure and mapping, as signaled in the M & M section, with no detectable contrast due to the presence of the polymer.  

Thank you for the clarification. Now the reviewer understands that line 185 “All topographic images were taken at 256 x 256-pixel resolution” refers to Figure 3a, 3c, and 5a only, and the mapping resolution is 20000nm / 30 on line 191. Nonetheless, please state the mapping grid 30x30 or 20x20 clearly in M&M.

Done (lines 206-207).

Thank you for the inclusion. But the reviewer made a mistake on previous comment. The reviewer would like to see the raw loading and unloading F-d curves (not the smoothened curves in Fig1a) from both native bone and callus. Please see the example from Fig2. on https://afm.oxinst.com/assets/uploads/products/asylum/documents/Nanomech-Pro-NanomechanicalAFM-Techniques-for-Diverse-Materials.pdf.

The curve shown in Fig. 1a is not smoothened. It is a 1000 values loading-unloading curve (this info added at line 207) and is taken from a real measurement; the inset of Fig. 1a just reports a zoom-in of the same curve, where noise is clearly visible. Nevertheless, noise raises clearly when the same curve it is converted into the corresponding F-h (Fig. 1b).

For the document provided, note that the curve in Fig. 2 spans on the y-axis one order of magnitude smaller than our values (microN), which may justify the slight noise that it features. The same applies to Fig. 4 of the same document. However, also the curve resolution matters, as well as the experimental context in which those curves were acquired.

Question: Percentile representation of data is quite inappropriate since the data obviously seems to follow lognormal distribution. If the authors want to show data heterogeneity, the reviewer recommends showing histogram distribution and quantified with mode or distribution fitting values.

Answer: Log-normal fittings on our data provided low (<0.8) r2; this circumstance (neither gaussian nor lognormal distributions of moduli) was already observed on cortical bone (Ref. 10) and is not surprising due to the heterogeneity of such samples. Thus, following the literature we reported percentile representation as more appropriate in this context. However, we remark that our purpose was not to provide exact estimations of the bone stiffness, but to develop a method to extract nanomechanical mapping from these surfaces.

Please include this discussion in the text.

Done (lines 341-343).

Reviewer 2 Report

All my comments have been properly addressed. All doubts were clarified in direct comments or by modification in the manuscript. The discussion of the article has been greatly improved. I recommend publication of the manuscript.

Author Response

We do thank the Reviewer for this decision.